# Evolutionary Toxicogenomics of the Striped Killifish (*Fundulus majalis*) in the New Bedford Harbor (Massachusetts, USA)

**DOI:** 10.3390/ijms20051129

**Published:** 2019-03-05

**Authors:** Paolo Ruggeri, Xiao Du, Douglas L. Crawford, Marjorie F. Oleksiak

**Affiliations:** 1Marine Biology and Ecology, Rosenstiel School of Marine and Atmospheric Science, University of Miami, 4600 Rickenbacker Causeway, Miami, FL 33149, USA; xdu@rsmas.miami.edu (X.D.); dcrawford@rsmas.miami.edu (D.L.C.); moleksiak@rsmas.miami.edu (M.F.O.); 2Dipartimento di Scienze della Vita e dell‘Ambiente, Universita‘ Politecnica delle Marche, Via Brecce Bianche, 60131 Ancona, Italy; 3Laboratory of Integrative Biology of Marine Models, CNRS-Sorbonne University, Station Biologique de Roscoff, Place Georges Teissier, 29680 Roscoff, France

**Keywords:** candidate outliers, migration, evolutionary genomics, GBS, SNPs, toxicant resistance

## Abstract

In this paper, we used a Genotyping-by-Sequencing (GBS) approach to find and genotype more than 4000 genome-wide SNPs (Single Nucleotide Polymorphisms) from striped killifish exposed to a variety of polychlorinated biphenyls (PCBs) and other aromatic pollutants in New Bedford Harbor (NBH, Massachusetts, USA). The aims of this study were to identify the genetic consequences of exposure to aquatic pollutants and detect genes that may be under selection. Low genetic diversity (*H*_E_ and π) was found in the site exposed to the highest pollution level, but the pattern of genetic diversity did not match the pollution levels. Extensive connectivity was detected among sampling sites, which suggests that balanced gene flow may explain the lack of genetic variation in response to pollution levels. Tests for selection identified 539 candidate outliers, but many of the candidate outliers were not shared among tests. Differences among test results likely reflect different test assumptions and the complex pollutant mixture. Potentially, selectively important loci are associated with 151 SNPs, and enrichment analysis suggests a likely involvement of these genes with pollutants that occur in NBH. This result suggests that selective processes at genes targeted by pollutants may be occurring, even at a small geographical scale, and may allow the local striped killifish to resist the high pollution levels.

## 1. Introduction

Anthropogenic stressors are a major threat to ecological balance in both terrestrial and aquatic environments. Most anthropogenic stressors are related to several classes of chemical compounds that are directly or indirectly released in the environment by human activities [1]. Although most of these chemicals are released in terrestrial environments, they affect marine environments because of the tight relationship between terrestrial and marine ecosystems [2]. For example, more than 80% of terrestrial contaminants reach coastal and oceanic waters [3]. These pollutants can negatively impact ecosystem functioning of both marine coastal and pelagic areas by reducing the primary production and increasing respiration [4], and these negative effects contribute to demographic instability of several marine species, which induces strong ecological shifts in species composition as well as physiological and evolutionary changes in wild populations [5,6]. The mass-mortality or the shift in available niches generated by some pollutants may alter natural patterns of genetic structure, which triggers selectively important micro-evolutionary events. Pollutants might act as a selective force, which eliminates individuals that cannot tolerate the pollutants and consequently reduces genetic variability [7,8].

The physiological and evolutionary consequences of several pollutants in wild populations are still not completely understood [9,10,11,12]. Finding pollutant-mediated evolutionary trajectories in aquatic populations is a complex task due to high local variability in pollutant mixtures, which can change the pattern and the target of genes affected by selection [13,14,15,16,17]. In this context, methods based on pools of pre-selected markers (i.e., a “candidate genes approach”) may not clarify how natural selection is acting when pollutant mixtures occur. The current Next Generation Sequencing (NGS) methods may offer a broader survey of genetic variation and an unbiased opportunity to scan population genomes and identify genome-wide polymorphisms affected by selective pressures from anthropogenic pollutants [10,18]. Among the NGS methods, the use of Genotyping by Sequencing (GBS) is currently becoming a fast, effective, and inexpensive choice to target thousands of single nucleotide polymorphisms (SNPs) throughout the genome [19] and characterize population genetic variation and genomic selection in aquatic toxicology [10].

A promising location to examine the population genetic effects of anthropogenic pollution is in New Bedford Harbor, Massachusetts, USA (hereafter NBH). NBH is a US Superfund site, a polluted site designated for cleanup with funds from the responsible party, and contains hazardous chemicals discharged into it since the mid-1940s. These hazardous chemicals pose a risk to humans and the environment needs to be cleaned up. Despite the cleanup effort in NBH, there is still a persistent pollution gradient that runs from the upper harbor (where the River Acushnet discharge freshwater) to the lower harbor. The pollution level was tracked by a monitoring program run between 1993 and 2009, and data about PCB levels were recorded [20,21]. The NBH hosts a complex pollution mixture made up of polychlorinated biphenyls (PCBs), polychlorinated di-benzo-*p*-dioxins, polychlorinated dibenzofurans, polycyclic aromatic hydrocarbons (PAHs), and metals [20,21]. These pollutants are associated with local biodiversity loss and affect fisheries and shellfish catch because tissues are too contaminated for public consumption [21]. Previous analyses have shown that exposure to environmentally-relevant concentrations of PCB-like compounds (i) have detrimental effects to fish population reproduction [22], (ii) induce embryological developmental anomalies [23], (iii) produce cardiotoxicity [24], and (iv) result in natural selection in wild fish populations [9,25,26].

Two common NBH fish species are the striped killifish (*Fundulus majalis*) and the Atlantic killifish (*Fundulus heteroclitus*) [27]. Atlantic killifish (*F. heteroclitus*) have been used to identify genetic and physiological responses to pollutants in the wild [23,25,28,29]. Key features of *F. heteroclitus* include: (i) large population sizes (>10,000 individuals), (ii) rapid tolerance to new environmental features, (iii) a wide range (that spans from New Brunswick, Canada to the North Atlantic Florida Coast), (iv) non-migratory behavior with limited seasonal movements (in the range of a few meters), and (v) short generation times (essential to evaluate if natural selection affects variation in allele frequencies across generations) [30]. The Atlantic killifish is able to respond physiologically to a large set of environmental variations and stress [11,28,31,32,33]. The *F. heteroclitus* population in the NBH was found to be tolerant to contaminants and showed genetic differences when related to congeneric populations in unpolluted sites [9,16,30,33,34,35]. For precisely the reasons mentioned above, *F. heteroclitus* is often used as a model to test physiological and evolutionary responses to local contaminants [9,25,36].

The striped killifish *F. majalis* shares similar biological and ecological features with *F. heteroclitus.* Both killifish species have a sympatric range, but, unlike the more well-known congener (*F. heteroclitus*), the striped killifish is (i) seldom found high in the upper-inner tidal, (ii) infrequently found in fresher water, and (iii) has a greater mobility, which allows this fish to move for several Km and have greater population connectivity [37].

Despite the little information about the evolutionary and physiological responses to pollutants, the striped killifish population in the NBH area persists, which suggests the possibility that it has developed resistance to pollutants. In addition, the striped killifish lacks many resources available for the Atlantic killifish (i.e., well characterized populations resistance to toxicants or extensive genomic resources) and is, therefore, a good model to assess the power of GBS methodologies and the effects of pollutants in the wild. The GBS approach has been infrequently used with non-model species in an ecotoxicological context and at a fine scale (in this study within a single harbor) [38]. The aim of this study is, therefore, to investigate the genetic variation in striped killifish samples collected along a steep pollution variation in the New Bedford Harbor (MA, USA). We apply an unbiased genomic approach (genotyping by sequencing, GBS) to simultaneously find and genotype thousands of SNPs in striped killifish and offer the possibility to add this species as a new model for genomic resistance to aquatic pollutants. We use multiple tests to detect candidate outliers and verify how they respond when used on non-model species exposed to steep variation in pollutants.

## 2. Results

### 2.1. NGS Sequencing

Illumina GBS sequencing using the TASSEL pipeline with the *Fundulus heteroclitus* reference genome (https://www.ncbi.nlm.nih.gov/genome/743) recovered 5403 SNPs found in at least 80% of individuals with 136 individuals having at least 70% of all SNPs. Among these SNPs, 1275 SNPs were in the Hardy Weinberg disequilibrium with observed heterozygosity significantly exceeding expected heterozygosity (HWE, *p* < 0.01). These SNPs were removed, and the final dataset of 4128 SNPs was subsequently used for statistical analyses.

The preliminary test for detecting SNPs under directional selection identified 564 SNPs as potential candidate outliers from all the six possible pairwise comparisons among sampling sites. Subsequent analyses, where neutral genetic variation was expected, were run using 2208 presumably neutral SNPs (excluding SNPs with significant linkage disequilibrium and candidate outliers).

### 2.2. Genetic Differentiation Between Populations and Gene Flow

Pairwise genetic differentiation (*F*_ST_ values) based on 2208 neutral SNP loci were all non-significant, which shows negative values that ranged between −0.0039 (PIL vs. FAH) and −0.0057 (HST vs. MAT, Table 1). When pairwise *F*_ST_ values were estimated using candidate outliers, 5 out of 6 pairwise population comparisons were significant and ranged between 0.0034 (PIL vs. HST) and 0.0303 (PIL vs. FAH, Table 1). The AMOVA test showed that most of the molecular variance was explained within populations (99.67%, FST: 0.0023) even though a significantly small portion was explained by groups (0.23%, FCT: 0.0032, Appendix A).

Results from DAPC were stable after retaining 28 out of 60 PCAs and showed the first two components as significant. In general, all four sampling sites were not well resolved. However, along the axis of the first significant component, a separation between PIL + FAH + HST against MAT appears (Figure 1). Similarly, along the axis of the second most significant component, a slight separation between PIL + FAH, MAT, and HST was detected (Figure 1).

STRUCTURE simulations of 2208 neutral SNPs yielded after the Evanno method, the most likely structure of two genetic clusters (K = 2) (Ln[P(K)] = −29,0230.24 ± 68.46) and a potential substructure for four genetic clusters (K = 4) (Figure 1, Appendix A). A similar structure (K = 2 and a substructure for K = 4) was detected using 1920 loci that included candidate outliers and loci under potential linkage disequilibrium (Figure 1, Appendix A). The simulation carried out using all SNPs yielded a higher log likelihood for three main genetic clusters (K = 3) (Ln[P(K)] = −531,638.50 ± 51.84). The same scenario was confirmed after the Evanno method (Appendix A). However, the q value distribution in both simulations did not detect a spatially explicit genetic structure among the four sampling sites (Figure 1).

Estimation of the number of first generation migrants showed large connectivity among sampling sites. Appendix A provides the percentage of individuals assigned to the same sampling site). Migration ranges between 8.82% in MAT and 16.67% in HST. The source of first-generation migrants was mostly HST (representing 45.71% in PIL, 35.48% in FAH, and 32.35% in MAT) and PIL (representing 38.89% in HST and 32.35% in MAT, Appendix A).

MIGRATE-n results showed values of the Θ parameter ranging from 0.00010 in MAT to 0.00057 in HST (Figure 2). The historical gene flow seemed to be symmetrical between every sampling location pair except between FAH and HST (M_FAH__→HST_ = 61.7, M_HST__→FAH_ =127.7, Figure 3). The M values ranged between 61.7 (M_FAH__→HST_) to 131.8 (M_HST-MAT_) (Figure 2).

### 2.3. Tests for Detecting Signatures of Selection

The *F*_ST_-based test to detect SNPs under directional selection in Lositan indicates 361 SNPs as candidate outliers. The pairwise comparison between PIL vs. MAT detected 117 candidate outliers (after 1% FDR correction), PIL vs. HST, 82 candidate outliers, FAH vs. MAT 115 candidate outliers, FAH vs. HST, 96 candidate outliers, and HST vs. MAT, 90 candidate outliers. 

Identification of outlier SNPs using a Hierarchical Island Method (HIM) implemented in Arlequin 3.5.2.2 detected 109 candidate outliers (after 1% FDR correction).

A third analysis to detect SNPs evolving by natural selection used Bayenv 2 analysis, which defines SNPs correlated with an environmental parameter (pollution) after correcting for demography [39]. A total of 128 SNP_S_ [with Log_10_(BF) > 1 and with empirical *p*-values < 0.05] were detected among the 10 replicates performed. Among them, 110 SNPs were detected only in one replicate, 13 SNPs were detected in two replicates, 3 SNPs were detected in three replicates, and 2 SNPs were found six times.

Outcomes from (i) Lositan, (ii) HIM, and (iii) Bayenv 2 revealed a total of 539 potential candidate SNP outliers. In all the tests performed, we detected outliers with a distribution across the whole *Fundulus* genome (Figure 3). There was overlap for three loci with all the three outlier’s detection methods (S0_4352665, S0_4352669, S9887_384963) including 56 SNPs were found to overlap between at least two detection methods (Lositan, Bayenv2 and HIM) (Figure 3).

### 2.4. Tests for Functional Annotation of Candidate Outliers

The sequences (75 base pair sequences) for the 539 candidate outliers were aligned against any available GenBank resource using the blastn algorithm. A total of 237 SNPs out of 539 (28.01%) had hits with significant (E-value < 0.0001) annotations. A total of 99.75% of those hits matched Eukaryote sequences. Additionally, 68.77% of the Eukaryote hits were related to teleost fish species (Appendix A) and 36.23% of them belonged to the Cyprinodontiformes Order, of which 41.76% of them specifically referred to *Fundulus heteroclitus*, which is the most closely related species to *F. majalis* with available genomic resources (Appendix A).

A total of 151 SNPs loci out of the 237 SNPs with annotations were related to coding regions (functionally annotated SNPs) NCBI ID codes for these functionally annotated SNPs, which were converted into human UNIPROT ID codes and used for the enrichment analysis in DAVID 6.7. These 151 SNPs produced a total of 958 Uniprot hits associated with 429 different human genes (*Homo sapiens*) (Appendix A). The analysis with DAVID 6.7 identified 44 different gene clusters and 14 out of 44 clusters showed significant associations (*p* < 0.05, Appendix A). These clusters are associated with 35 functional pathways and 25 out of 35 pathways showed significant cellular/biochemical/physiological functions (Bonferroni *p* -values < 0.05, Table 2). A total of 1126 of the total 1147 Uniprot hits were also significantly (*p* -value < 0.001) related to 13 different disease classes that span from metabolic (167), cardiovascular (137), and cancer (103) to developmental (60) and reproductive (37) pathologies (Table 3).

## 3. Discussion

Evolutionary toxicology, which is the study of evolutionary responses of populations to human-mediated stressors, is still in an early stage [40]. It aims to understand how pollutants contribute to (i) changes in genome-wide genetic diversity, (ii) population differentiation, and (iii) local selection at genes that modify physiological pathways associated with pollutant resistance [40]. Genomic approaches provide unbiased genome-wide analyses that help clarify populations’ molecular evolutionary responses to pollutants.

In this study, we investigated how natural selection affects local population exposed to contaminants in an estuarine, non-model species: the striped killifish (*F. majalis*). Particularly, this method successfully allowed us to obtain more than 4000 SNP markers from *Fundulus majalis*, that were used to detect allelic shifts in loci potentially connected to the exposure to different levels of pollutants. Similar approaches were previously tested comparing genomes of individuals collected from contrasting environments [12,41,42]. This strategy should help define loci under strong selection, typically in isolated environments. An important caveat is that we are assuming that pollutants, primary PCBs, are important selective forces. Yet, other abiotic (salinity) and biotic (community composition) factors could also co-vary with PCBs and may also be selectively important. The focus on pollution in this study seems reasonable because we are investigating a single population with samples separated by few Km of coastal area where the pollutant concentration has the largest environmental variation relative to many other abiotic factors. Additionally, this approach is supported by the observation that more than half of the annotated outlier loci were related to genes known to be influenced by PCBs. The factors include (a) regulation of apoptosis and cell lifespan, (b) cancer, and (c) hormones and neuropeptides signaling. In this case, we examine the genetic variation of a population of *Fundulus majalis* in New Bedford Harbor (Massachusetts, USA) exposed to a large variation in aquatic contaminants.

### 3.1. Genetic Variability and Population Differentiation for F. majalis in the New Bedford Harbor 

Along the pollution gradient, our data indicate consistently high genetic variation. This goes against basic assumptions in evolutionary toxicology and the ‘‘genetic erosion hypothesis’’ postulated by van Straalen and Timmermans [43] that suggests genetic variability erosion due to rapid demographic collapse when populations are faced with contaminants. Such an event can decrease genetic variability promoted by genetic drift and selection [40]. This model predicts that “genetic erosion” should be proportional to the proximity of individuals to the contaminant source and should result in a progressive loss of genetic variation with lower diversity in the most polluted sites and higher diversity in the unpolluted ones. Yet, in this study, there is no apparent progressive reduction in genetic variation (Figure 4, expected heterozygosity and nucleotide diversity) along the pollution gradient. The same result was observed in a genotyping-by-sequencing analysis for a rove beetle (*Staphylinus erythropterus*) population along a heavy metals pollution gradient in Poland where there was no clear pattern of reduced genetic variation measured from over three thousand SNPs in beetles located close to the pollution source [38]. This beetle study also found weak genetic differentiation between sampling sites and suggested that it was due to extensive gene flow among sites. Therefore, high beetle mobility may counteract genetic variation loss associated with pollution by continuous migration from nearby, less impacted sites. Similar reasoning could explain the data for the NBH striped killifish population. A lack of population structure in the striped killifish collected in the NBH area and high connectivity among sampling sites were detected. This connectivity is likely related to the mobility of the species, which covers similar distances to those that separate the sampled sites (<20 Km). Boorse and Storlie [44] suggest that striped killifish daily movements can enhance the connectivity of local groups of individuals. This observation suggests that connectivity within populations may contribute to counteract the local genetic variability loss [45,46]. Yet, even in the face of high connectivity, our data suggest that evolutionary divergence affects allele frequencies at many SNP loci. This is in agreement with new theories that suggest the need for high genetic variation in order to maintain adaptive responses in populations chronically exposed to stressful environments [47]. In this context, selective divergence with high gene flow could be advantageous by increasing population size and introducing new alleles that may be locally beneficial [46,47]. The evidence for and the problems with identifying selective divergence in the striped killifish are discussed below.

### 3.2. What Tests for Candidate Outliers can Tell Us about Selective Response to Pollutants: A Matter of Relative Performance

One of the main challenges in ecological genomics is to identify genes underlying functional traits, such as toxicant resistance [18,48]. Currently, tests to detect selectively important genetic variation are based on different assumptions that affect their statistical power and accuracy [49,50,51]. In general, comparing the results from several tests is a good practice to assess the occurrence of functionally relevant genetic variation [49,52]. In this study, we used three different tests for detecting outlier loci based on (i) *F*_ST_-based pairwise comparisons of most divergent loci, (ii) an *F*_ST_-based approach that can account for a hierarchical structure, and (iii) a Bayesian correlation test between pollutant concentrations and the genetic variation at loci. The proportions of candidate outliers ranged between 2.64% and 8.75% of the total SNPs markers we retrieved. These percentages are roughly in accordance with the relative number of SNPs showing significant genetic variation (<5–10% of loci screened) found in other studies [48,49,50,51,52,53].

There was limited overlap among the methods applied (Lositan, HIM and Bayenv2) with only three overlapping loci. Limited overlap is a frequent issue when comparing genome-environment association methods (i.e., Bayenv2) with others that rely on divergence in *F*_ST_ (i.e., Lositan and HIM) [49,50,51] and depends on the assumptions made by the different statistical tools that could not perform in the same manner when dealing with (i) polygenic traits, (ii) panmictic/highly connected populations, and/or (iii) when the selective agent is actually represented by a mix of multiple stressors/features.

For instance, several investigations demonstrated that the polygenic nature of phenotypic traits poorly accord with the basic assumptions of most genome-wide scan approaches [12,54,55]. With polygenic selection, allele frequency changes only need to occur at a subset of loci that could alter an evolutionary trait. Thus, signals of selection are distributed across many loci and are not necessarily the same in all individuals, which leads to weak individual-locus detection for selection [54,55]. It could be supposed that genes functionally relevant for tolerance to pollutants might be polygenic traits and, hence, vary in the degree of detectability by the different methods used in this scenario.

Demography also affects the power of outlier tests producing a high type II error (falsely accepting the null hypothesis) [49,50,51]. This is frequently found when there is a lack of population structure or in the case of strong connectivity [56]. In this context, *F*_ST_-related estimates between sampling sites would be lowered by higher gene flow and, in turn, affect the power of tests to detect loci under selection [49,57]. A similar outcome was proposed in a study on the panmictic American eel (*Anguilla rostrata*) that also exhibited less than 1% of candidate outliers from a genome-wide scan performed at a larger spatial scale than this study [58]. This scenario fits well with the high level of connectivity we detected in *F. majalis* from NBH and could be an explanation for the small overlap of candidate outliers obtained.

Similarly, the effects generated by a gradual variation of the agent that is supposed to generate local selection can lead to analogous outcomes. In fact, other studies have pointed out how environmental gradients break the assumption of population isolation and exposure and lead the genome scan methods to bias the expected number of outlier loci [56,57].

The role of multiple interacting stressors could also lead to variation in the relative performance produced by different tests to detect outliers. The New Bedford Harbor (NBH) pollution source comes mainly from PCBs, but they are not the exclusive contaminants in the area [20,21].

Although we assumed that all contaminants in NBH follow the same pattern of concentration, we can speculate either that contaminants other than PCBs (to which information about local concentrations are not available) could be more greatly affecting (i) the selective responses in the striped killifish or (ii) the responsiveness of the environmental correlation methods employed (i.e., Bayenv2).

### 3.3. Genome-Environment Interaction in F. majalis from NBH

Although the outliers SNPs obtained from four different methods did not consistently overlap, enrichment analysis of the functionally annotated SNPs suggest the possibility that these loci relate to important traits that respond to pollution in the striped killifish. The enrichment analysis showed that most of these markers can be clustered in 14 different functional groups involved in: (i) antioxidant, xenobiotic, and fatty acid metabolisms, (ii) control of the cell fate (i.e., apoptosis, cell proliferation), (iii) control of the innate immunity, and (iv) neuromodulation. Similarly, these genetic clusters reflect 25 functional pathways that control analogous physiological responses. Most of the functionally annotated SNPs were also related to pathologies that span from cancer, cardiovascular, metabolic, and neurological to reproductive diseases. All these disease categories are known to be associated with physiological cellular interactions with xenobiotics, such as PCBs and other aromatic compounds [59,60,61]. These toxicants are usually highly lipophilic, and their functions are connected to membranes and the vesicles structure as well as to their transmembrane molecular crosstalk [61]. 

Several targeted SNPs are in genes involved in ATP-GTP binding activities (i.e., Ras-related protein RAB38 and syntrophin alpha) or with calcium-dependent molecules (i.e., susd1, cacng5, PLCD4, and stx2). These genes can be connected with the cardiovascular activities, and, in general, the cardiovascular system has been identified as a main target of dioxin-like compounds toxicity in all vertebrates [61] and more specifically in *F. heteroclitus* [62]. One of the calcium-dependent genes, syntaxin 2, was also identified to be involved in the acrosomal reaction and with sperm mobility [63]. This function may be put in relation with the estrogen-mimicking behavior that many xenobiotics (i.e., PCBs) assume and their role as endocrine disruptors for many organisms, including fish [22,64]. PCBs act as estrogen-like molecules in both males and females, inducing cancer and reproductive alterations, like induced feminization in males [65]. The reproductive consequences of PCBs are supported by the significant allele frequency change for the estrogen receptor 1 (esr1). This gene receptor is one of the main targets for synthetic xenoestrogens [22,66] and it plays a protective role from PCB-induced DNA damage in human breast cancer cells and apoptotic cycle alterations [67]. It also has been shown to be differentially regulated in NBH *F. heteroclitus* compared to reference fish [68,69]. 

Additionally, our approach identified genes directly connected with the metabolism of xenobiotic like the cytochrome P450 genes (CYP2F2, CYP2K4, CYP2P2) and the Aryl Hydrocarbon Receptor Nuclear Translocator 2 (ARNT2). A total of 137 genes encoding P450s were recently identified in fish, and their general activities relate to detoxifying systems against pollutants [70]. CYP2F2 and CYP2K4 are specifically associated with epoxygenase and hydroxylase activities of arachidonic acid, which is a polyunsaturated fatty acid involved with cellular signaling and inflammatory responses [71]. A significant transcriptional alteration of a member of the CYP2 family (CYP2N2), which is also involved with molecular mobilization of arachidonic acid, was also detected in the NBH *F. heteroclitus* population where the lower CYP2 expression in the polluted populations could be related to a limitation in energy storage and other metabolic functions [31]. The change in CYP2 allele frequency could have a similar role for *F. majalis* from the same area. The Aryl Hydrocarbon Receptor Nuclear Translocator 2 (ARNT2) was also demonstrated to be one of the genes that transcriptionally respond to exposure from PCBs and other aromatic compounds in *F. heteroclitus* embryos [72]. This gene resists pollutants developed by *F. heteroclitus* in the NBH, and it can be speculatively proposes that it has the same phenotype results for the *F. majalis* population we examined.

Furthermore, we detected SNPs loci associated with metallothioneins (MTs), which is a group of functionally diversified proteins with metal binding and redox capabilities that confer tolerance to metal pollutants. An increase in MTs is frequently associated with hepatotoxicity due to PCB exposure in fish. MTs could be similarly involved in toxicant resistance of *F. majalis* in NBH.

Another interesting result comes from the cluster that contains the Leucine-rich repeat genes (LRRs, genomic components that are involved in the innate immune response) and the functional pathways connected with the immunological response to bacterial infections. In fact, it is not so surprising that the existing connection between chronic exposure to contaminants like PCBs and metals and the development of transgenerational immunological suppression effects. Nacci et al. found that both adults and embryo *F. heteroclitus* from NBH are producing an immune response comparable to control specimens, and, therefore, they proposed that this killifish population might have evolved mechanisms that minimize the immuno-suppressive effects of PCBs [73]. A similar consideration could be proposed for the striped killifish population that we examined. The emerging genes from our GBS analysis that relate to the immunological response to bacterial infections could suggest that *F. majalis* from NBH have developed a genomic resistance that contrasts the immunosuppressive effect of PCBs or that, in general, genes related to the immune system are playing a role in the NBH population.

These findings suggest that local selection of genes that resist toxicants may occur in *F. majalis*. However, other relevant genes found in the adaptation of *F. heteroclitus* to PCB-contaminated environments [35,62] were not identified. In particular, we found no association between nucleotide variation and PCB-tolerant phenotype at the aryl hydrocarbon receptors 1 and 2 (AHR1, AHR2) [35,62], cathepsin Z, the cytochrome P450s (CYP1A and CYP3A30), and the NADH dehydrogenase subunits genes [62]. This points out one of the short-comings of GBS. GBS samples of approximately 0.1% of the *Fundulus* genome often does not assay all relevant genes. Thus, although we observed potential selection acting on previously described genes (i.e., the estrogen receptor 1 and CYP450 genes), the GBS approach is not comprehensive in detecting all loci implicated with *F. majalis’* selective response to local pollutants.

## 4. Materials and Methods

One-hundred and fifty-eight striped killifish were collected from the New Bedford Harbor (NBH, MA, USA). These individuals were collected in four sampling sites representative of a polychlorinated *biphenyl* (PCB) gradient measured as standard dry mass weight of the sediment, as reported in Roark et al. [36]. Two collection sites were in the Acushnet River estuary within New Bedford Harbor (Figure 4, PIL and FAH). These sites represent the areas closest to the pollution source where striped killifish naturally can occur. The other two collection sites have lower pollution levels. One site was in the New Bedford Outer Harbor (Figure 4, HST) while the most distant sampling site was in the Mattapoisett Harbor (Figure 4, MAT). The latter site was defined as a reference site for our analysis. This is because it: (i) experiences lower pollution levels, (ii) belongs to a different embayment system than NBH, and (iii) represents the farthest geographical site from the pollution source compared to the other sampled sites. Fish were collected using minnow traps. All fish were returned to their environment after removal of small fin clips (<10 mm^2^). Fin clips were stored in 320 μL of Chaos buffer (4.5 M guanadinium thiocynate, 2% N-lauroylsarcosine, 50 mM EDTA, 25 mM Tris-HCL pH 7.5, 0.2% antifoam, 0.1M β-mercaptoethanol) at 4 °C prior to being processed. Genomic DNA was isolated using a silica column protocol [74] and the DNA quality was assessed via agarose gel electrophoresis. DNA concentrations were quantified and standardized using AccuBlue broad range quantitation assays, according to the manufacturer’s instructions (Biotium Inc., Fremont, CA, USA; https://biotium.com/).

The GBS library was prepared as described in Elshire et al. [19] using the restriction enzyme *Ase* I. Adaptors (0.4 pmol/sample) were ligated to 50 ng of gDNA. The GBS library was sequenced on a single lane using Illumina HiSeq 2500 with a 75 bp single-end reads (Elim Biopharmaceuticals, Inc., Hayward, CA, USA; https://www.elimbio.com/). Sequence reads were aligned and putative SNPs were retrieved using the TASSEL 5.0 pipeline [75] with the *F. heteroclitus* genome as a reference [76]; https://www.ncbi.nlm.nih.gov/genome/743]. Discovered SNPs were filtered with TASSEL 5.0 retaining individuals with at least 70% of SNPs and SNPs found in at least 80% of individuals. Allele frequencies and heterozygosities (*H*_E_, *H*_O_) from filtered data were calculated per sampling site and per locus using Arlequin 3.5.2.2 [77]. SNPs with observed heterozygosity (*H*_O_) exceeding expected heterozygosity (*H*_E_, *H*_O_ > *H*_E_) and significantly different from Hardy-Weinberg expectation (*p* < 0.01) were excluded. This latter filter is used to remove potential SNPs that represent differences between paralogs versus true allelic variants for a single locus [78,79]. Overall and by-population estimates of nucleotide diversity, (π) were computed using DNAsp v5 [80]. Initially, SNPs under neutral and non-neutral expectations were evaluated with Lositan [81]. Lositan settings consisted of 500,000 iterations based on an Infinite Allele model. A list of candidate outliers detected under positive selection was annotated from each possible pairwise comparison. Each test between localities (for a total of six pairwise comparisons) was replicated three times, and the final list of candidate outliers (presumably non-neutral markers) was obtained by merging lists from the three replicates. A False Discovery Rate (FDR, 1%) correction method was applied after evaluating significances with *p*-values < 0.01. This strategy was applied with the aim to use a more stringent criterion to detect significant, non-neutral SNPs.

The population structure was evaluated using AMOVA in Arlequin [77], multivariate discrimination analysis (DAPC) in the R package ADEGENET 1.3-4 [82] (http://adegenet.r-forge.r-project.org/), and STRUCTURE version 2.3.4 [83] (https://web.stanford.edu/group/pritchardlab/structure_software/release_versions/v2.3.4/html/structure.html). Migration rates were evaluated as implemented in GENECLASS 2.0 [84] as well as MIGRATE-n v. 3.5.1 [85] (https://peterbeerli.com/migrate-html5/index.html). The genetic distance among populations was estimated using *F*_ST_ values [86], as implemented in Arlequin [77] (http://cmpg.unibe.ch/software/arlequin35/Arl35Downloads.html). Among the four sampling sites, the variation within and among populations was determined by analyzing molecular variance (AMOVA, Arlequin) [77]. In the AMOVA, a hierarchical structure based on both the PCB concentration and geographical distance (measured as the shortest linear sea distance among sampling sites) was provided. The two most polluted sites were grouped together in the first cluster (PIL and FAH) while the remaining two sampling sites (HST and MAT) represented two other distinct clusters. A thousand permutations were applied to test the significance of molecular variance partitions, and a threshold of 1% (*p*-value < 0.01) was assumed.

Discriminant analysis of principal components was performed using the software DAPC [82] implemented in the ADEGENET 1.3-4 package [87] since R DAPC is a multivariate approach that describes the genetic differences among groups while minimizing the differences within groups using principal components of genetic variation (PCA). A set of 60 principal components was retained as predictors for discriminant analysis and cross-validation using the default setting was performed.

A Bayesian method performed with STRUCTURE version 2.3.4 [83] was used to cluster individuals and investigate the population structure. An admixture model and correlated allele frequencies were assumed. Simulation settings were based on 10,000 discarded iterations (burn-in) and 50,000 MCMC (Monte Carlo Marcov Chain) retained replicates. The population structures were comprised of between one and six clusters (K) were simulated, and each K was tested five times. Two separate simulations were performed based respectively on (i) exclusively neutral and (ii) all markers. The online software STRUCTURE HARVESTER 0.6.92 [88] (http://taylor0.biology.ucla.edu/structureHarvester/) was used to evaluate results under the Evanno method expectations [89] while CLUMPAK server (http://clumpak.tau.ac.il) [90] was used to obtain graphical plots.

Gene flow and connectivity between sampling sites were evaluated by two separate methods using the same 2208 presumably neutral SNPs (as tested from Lositan runs in a pre-screening trial). The first method allowed the calculation of current gene flow levels by estimating the number of first generation migrants using the Bayesian method of Rannala and Mountain [91] implemented in GENECLASS 2.0 [57] (http://www1.montpellier.inra.fr/URLB/GeneClass2/Help/). A set of 1000 MCMC iterations was produced to test the most probable site of each individual’s origin based on their multi-locus neutral genotype assets. A threshold *p*-value of < 0.01 was used to assess the significance.

The second method to detect gene flow and connectivity was based on the coalescent calculation of historical migration rates between sampling locations using MIGRATE-n v. 3.5.1 [85]. A Bayesian method was applied [85] and *F*_ST_ estimates among localities were used as a baseline for calculating the demographic parameter Θ (Θ = 4*N*_e_μ, where *N*_e_ is the effective population size and μ is the mutation rate) and the historical migration rate, M (M = m/μ, where m is the immigration rate per generation and μ is the mutation rate). Negative *F*_ST_ were set to zero. A Brownian motion model was used, and mutation was considered constant over a set of 1000 neutral SNPs. The MCMC procedure consisted of one long chain with 500,000 recorded genealogies for each locus, with 10,000 genealogies discarded as the burn-in.

Two *F*_ST_-based tests were performed to detect SNPs under positive selection. The first *F*_ST_-based test was carried out using Lositan [81] (https://popgen.net/soft/lositan/), which extracted potentially selected SNPs from pairwise comparisons (PIL vs. MAT, PIL vs. HST, FAH vs. MAT, FAH vs. HST, and HST vs. MAT) of PCB polluted sites in NBH against the reference sampling site in Mattapoisett Harbor (MAT). The Lositan settings were the same used for the preliminary analysis in which neutral and non-neutral SNPs were identified. SNPs under potentially positive natural selection (candidate outliers) were considered only if SNPs were significant in every replicate. The second *F*_ST_-based test was carried out in Arlequin [77,92] using the Hierarchical Island method (HIM). This method differs from the Lositan test by evaluating the impact of a hierarchical structure among the collected samples. The hierarchical structure considered in this test took into account site-by-site variation in PCBs. The first cluster included samples that are experiencing high pollution (PIL and FAH), the second cluster included a sample (HST) with low PCB concentrations, and the third cluster was represented by the reference samples collected in the Mattapoisett Harbor (MAT). HIM settings were characterized by simulating 100 demes with 10,000 coalescent iterations under the assumption of the hierarchical structure mentioned above. A significance threshold of 1% (*p*-value < 0.01) and a 1% FDR were applied to increase the level of accuracy in avoiding false positive tests.

The last test to identify candidate outliers was based on a statistical procedure in Bayenv2 [39] (https://gcbias.org/bayenv/). This method assesses genomic evidence for natural selection to specific environments by performing locus-by-locus Bayesian analysis to detect linear relationships between allele frequencies and environmental variables, assuming non-linear allele frequency changes [39]. The variance-covariance matrix was built up using 500,000 MCMC iterations in a simulation that involved only presumed neutral SNPs (candidate outliers and linked SNPs were excluded from the dataset). In order to apply a more rigorous approach, the last 200 variance-covariance matrices were averaged, and the resulting matrix was used to test environmental correlation against all SNPs (4128). A set of 10 replicates of environmental correlation were produced, and logarithmic values of Bayesian Factors (BF) were averaged in each locus. Only SNPs that showed average logarithmic values of Bayesian Factors (BF) larger than one (Log_10_BF > 1) were considered significant and potentially evolving by natural selection.

An intersection analysis was performed to find candidate outliers jointly identified by all these methods. A Venn diagram, that represents the candidate outlier’s intersections, was produced.

To annotate potentially selectively important genes, a FASTA file containing candidate outlier sequences was loaded in the online software BLAST (http://blast.ncbi.nlm.nih.gov/Blast.cgi?PROGRAM=blastnandPAGE_TYPE=BlastSearchandLINK_LOC=blasthome) [93] good matches (E-values < 0.0001) between the queries and any similar sequence in GenBank were searched using the *blastn* algorithm. The annotated sequences from candidate outliers were further used for a UniProt (http://www.uniprot.org) search. The search was aimed to find Human UniProt code associated with candidate outliers found in *Fundulus* and other organisms’ annotations. These UniProt codes were subsequently used in DAVID 6.7 [94] (https://david.ncifcrf.gov/content.jsp?file=citation.htm) to produce Functional Annotation Clustering and provide further information about the molecular and biological pathways in which annotated genes are involved.

## 5. Conclusions

We found that samples collected within a radius of 20 km (shoreline distance) in New Bedford Harbor experiencing a steep variation in aquatic pollution, exhibited ~530 SNPs with unexpected genetic variation patterns. The high mobility and connectivity within this striped killifish population made it difficult to identify changes in overall genetic variation (*H*_E_ or π) associated with the role of pollutants. The enrichment analysis of loci likely under selective processes revealed that most of these SNPs occur in genes that are presumably associated with pollution resistance and are involved in physiological processes that can determine an inherited tolerance of this population to pollutants. The unbiased approach of GBS that detected functionally-relevant loci suggests that, even in a species with large migration rates, pollution can significantly affect allele frequencies. Yet, the lack of agreement among different tests for detecting SNPs under selection might suggest how these tests are sensitive to different evolutionary forces and/or might lack the power to identify important loci (type II error) or mis-identify genes (type I error). In addition, there is the possibility that other environmental features may have reduced the power of the genomic scan analyses, which limits the ability to detect more loci associated with pollutant resistance. Overall, the evidence still suggests non-neutral divergence among samples that occurs in spite of a small geographic scale, a panmictic population, and large migration rates.

## Figures and Tables

**Figure 1 ijms-20-01129-f001:**
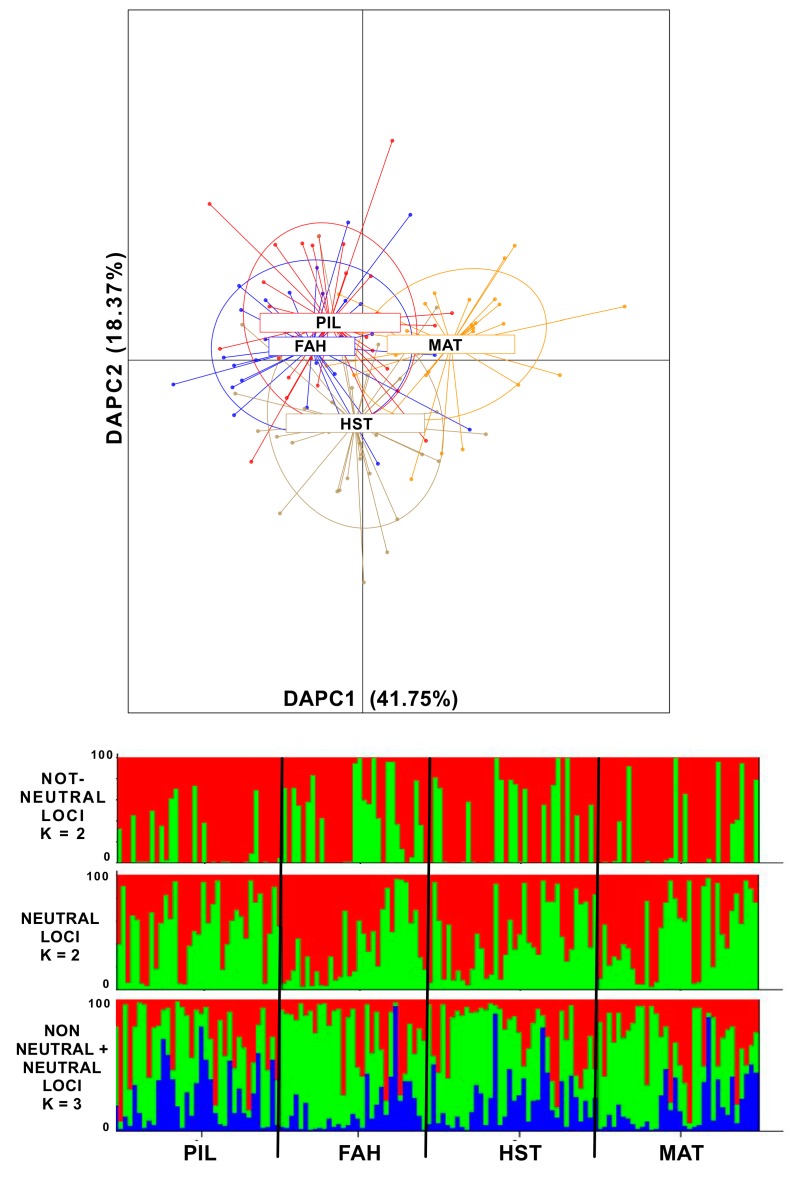
Figure 1 (above) DAPC (Discriminant Analysis of Principal Components) plot of individuals from the four sampling sites along the two most significant axes (PC1 41.75%, PC2 18.37% explained variance). In red are the individuals from Pilgrim Avenue (PIL) and, in blue, are the individuals from Fairhaven (FAH). Both PIL and FAH are located in the inner New Bedford Harbor (NBH) close to the pollution source. Depicted in light brown are individuals from Hacker Street (HST) and, in yellow, are individuals from Mattapoisett (MAT). Both these latter sampling sites are located in the outer NBH area and are exposed to lower pollution concentrations. The insert explains the percent of variance for the first three discrimination eigenvalues (in dark grey are the significant DAs). Figure 1 (below), STRUCTURE plots of the most likely K tested from the simulations with non-neutral (K = 2), only neutral (K = 2), and the complete set of markers analyzed (K = 3).

**Figure 2 ijms-20-01129-f002:**
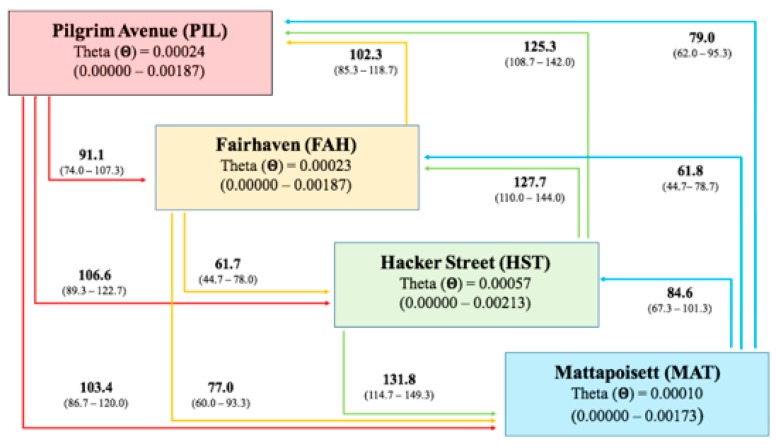
Pattern of historical gene flow from the Migrate-n simulation. The squares represent the four sampling locations and theta (Θ) values with 95% confidence intervals. Colored arrows refer to the sampling location of the same color and explain the directionality and rate of gene flow (M values are reported at the corner of each arrow with a 95% confidence interval).

**Figure 3 ijms-20-01129-f003:**
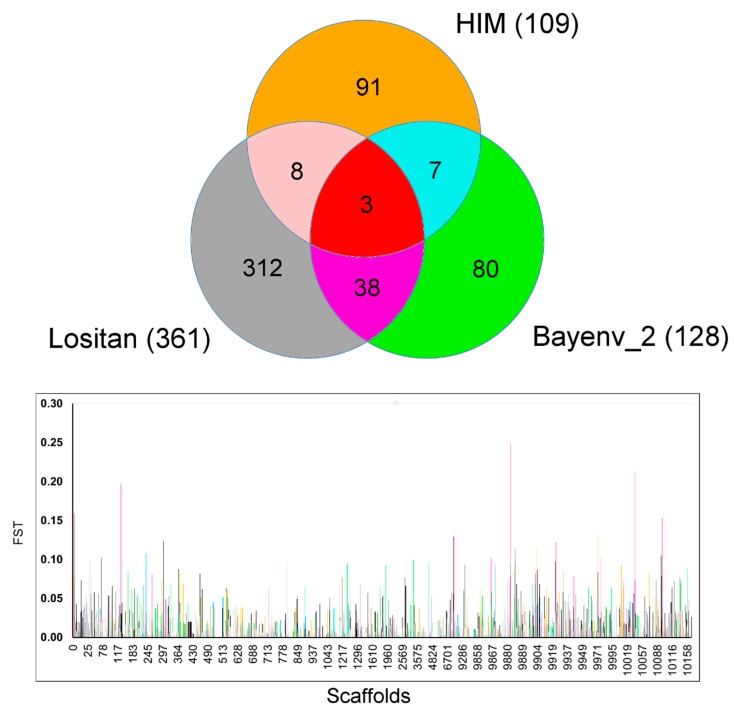
Venn diagram of the outlier loci interpolation among four candidate outlier detection tests and *F*_ST_ value distribution of each outlier locus along the scaffolds position. In the Venn diagram (above), the numbers encased represent the number of shared SNPs between and among the corresponding set of methodologies. The colors in the Venn diagram reflect the pattern of color used in the Manhattan plot (below) that represents the scaffold position of each outlier locus and its estimated *F*_ST_ value.

**Figure 4 ijms-20-01129-f004:**
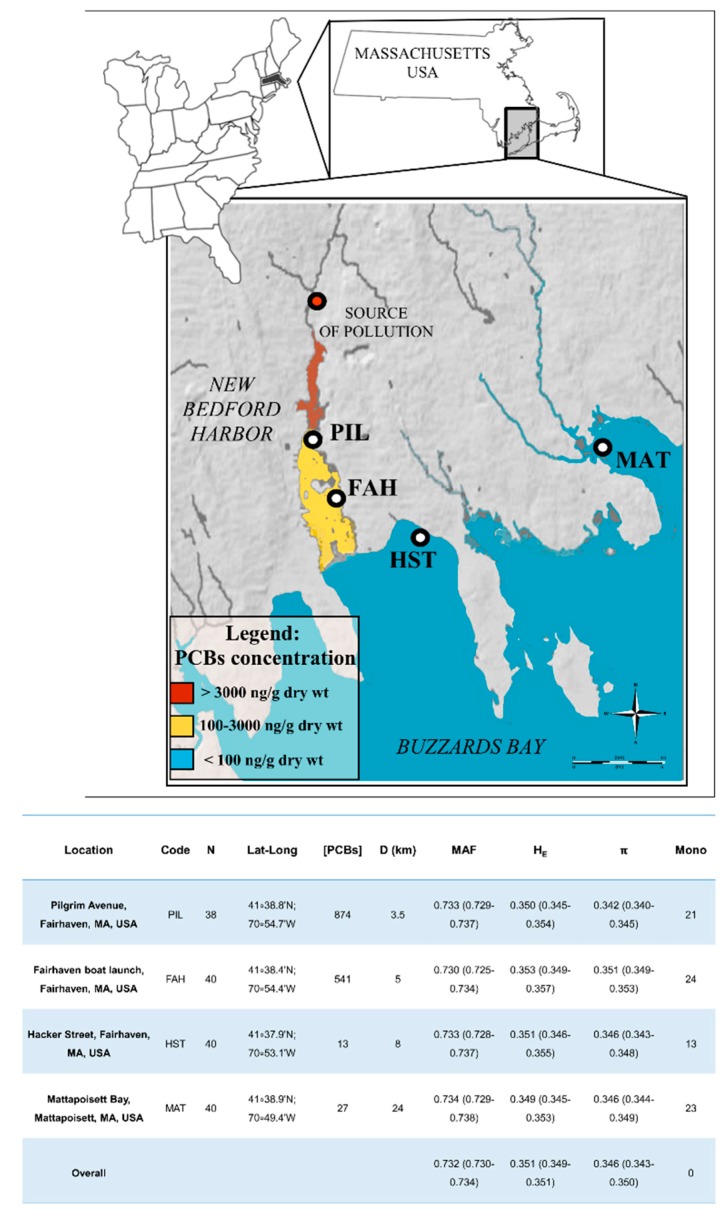
Geographical area and the sampling locations covered by this study. The main map refers to New Bedford Harbor and Buzzards Bay (URL: http://mapmaker.education.nationalgeographic.org). The colored coastal areas refer to the pollution gradient described in the figure legend. Map ruler (URL: https://upload.wikimedia.org/wikipedia/commons/1/1a/Scale_kilometres_miles.svg) and compass (URL: https://commons.wikimedia.org/wiki/File:Simple_compass_rose.svg). The map of Eastern USA was obtained from URL: http://www.clker.com/cliparts/m/3/H/f/K/e/eastern-u-s-map-md.png). The map representing the Massachusetts outline was downloaded from Worldatlas.com at the URL: http://www.worldatlas.com/webimage/countrys/namerica/usstates/outline/ma.htm). The table below the figure summarize the main pollution and genetic diversity observed. Code = collection site codes, N = sample size, Lat-Long = geographic coordinates of sampled points, [PCBs] = PCBs concentration in (ng/g dry weight), D = distance in Km from the source of pollution. Genetic diversity by population: averaged Major Allele Frequencies (MAF), averaged expected heterozygosity (*H*_E_), and averaged nucleotide diversity (π), and number of monomorphic loci (Mono). Values between brackets refer to 95% confidence intervals.

**Table 1 ijms-20-01129-t001:** Pairwise F_ST_ values between sampling sites based on neutral genetic markers (2208 SNPs, below the diagonal) and non-neutral genetic markers (linked loci + candidate outliers, above the diagonal). * *p*-value < 0.05, ** *p* -value < 0.01, *** *p* -value < 0.001.

	PIL	FAH	HST	MAT
PIL	0	0.0303 **	0.0034	0.0111 *
FAH	−0.0039	0	0.0250 ***	0.0221 ***
HST	−0.0048	−0.0053	0	0.0158 **
MAT	−0.004	−0.0052	−0.0057	0

**Table 2 ijms-20-01129-t002:** The list of 35 cellular/physiological pathways targeted by the enrichment analysis (KEGG Pathways) with DAVID 6.7. The analysis includes results from a list of 429 genes. N = number of genes participating to the KEGG Pathway. % = percentage of genes in the total of 429 genes. Category = represents the general cellular/physiological function to which the KEGG pathway is involved in. The Kegg pathway terms in red are not supported by significant P-values (P > 0.05). Categories: a = Metabolism. b = Cellular differentiation/survival. c = Cellular organization/adhesion. d = Cancer. e = Development. f = Immune response. g = Reproduction. h = Inflammatory processes. I = Neuronal transmission.

KEGG Pathway Terms	N	%	*p*-Value	Category
Nitrogen metabolism	11	2.5	3.40 × 10^−12^	a
PPAR signaling pathway	11	2.5	1.30 × 10^−5^	a
Mineral absorption	9	2.1	2.70 × 10^−5^	a
Neurotrophin signaling pathway	14	3.2	2.40 × 10^−5^	b
Regulation of actin cytoskeleton	18	4.2	6.30 × 10^−5^	c
ErbB signaling pathway	11	2.5	1.30 × 10^−4^	b, d
Adherens junction	8	1.9	3.30 × 10^−3^	c
Ras signaling pathway	15	3.5	3.80 × 10^−3^	b, c
MAPK signaling pathway	16	3.7	4.20 × 10^−3^	b, h
Fc gamma R-mediated phagocytosis	8	1.9	8.30 × 10^−3^	b, c, f
Proteoglycans in cancer	13	3	9.20 × 10^−3^	b, c, d
Tight junction	8	1.9	1.00 × 10^−2^	b, c
Epithelial cell signaling in *Helicobacter pylori* infection	7	1.6	1.00 × 10^−2^	b, f, h
Pathogenic *Escherichia coli* infection	6	1.4	1.30 × 10^−2^	c, f, h
Rap1 signaling pathway	13	3	1.30 × 10^−2^	b, c
Leukocyte transendothelial migration	9	2.1	1.40 × 10^−2^	c, f, h
Endocytosis	14	3.2	1.60 × 10^−2^	a, c
cAMP signaling pathway	12	2.8	2.10 × 10^−2^	b, c, e, f, g
*Salmonella* infection	7	1.5	4.60 × 10^−2^	b, f, h
Renal cell carcinoma	6	1.4	3.50 × 10^−2^	b, d
Arhythmogenic right ventricular cardiomyopathy (ARVC)	6	1.4	4.20 × 10^−2^	c, f, h
Cell adhesion molecules (CAMs)	9	2.1	4.20 × 10^−2^	c, f
Inflammatory mediator regulation of TRP channels	7	1.6	4.30 × 10^−2^	h
Estrogen signaling pathway	7	1.6	4.60 × 10^−3^	b, g
T cell receptor signaling pathway	7	1.6	5.80 × 10^−2^	b, f, h
Choline metabolism in cancer	7	1.6	6.00 × 10^−2^	b, c, h
Dopaminergic synapse	8	1.9	6.40 × 10^−2^	i
Bacterial invasion of epithelial cells	6	1.4	6.40 × 10^−2^	c, f, h
Non-small cell lung cancer	5	1.8	6.80 × 10^−2^	b, d
Regulation of lipolysis in adipocytes	5	1.8	5.80 × 10^−2^	a
Hepatitis C	8	1.9	7.50 × 10^−2^	d, h
Apoptosis	5	1.2	9.10 × 10^−2^	b, d
Shigellosis	5	1.2	1.00 × 10^−1^	d, h
Central carbon metabolism in cancer	5	1.2	1.00 × 10^−1^	d, h

**Table 3 ijms-20-01129-t003:** The list of 13 group of diseases targeted by the enrichment analysis (DAVID 6.7). The analysis includes two tests to assess significance of the genes participating to each disease category (Permutation test P-value and FDR correction). N = number of genes with implication in the categorized disease. % = percentage of representation in the total of 429 genes.

Disease Category	N	%	P-Value	FDR
Metabolic	167	34.9	1.80 × 10^−3^	4.60 × 10^−3^
Cardiovascular	137	28.7	4.50 × 10^−3^	1.00 × 10^−2^
Chemo-dependency	115	24.1	1.90 × 10^−2^	2.80 × 10^−2^
Pharmacogenomic	109	22.8	2.40 × 10^−6^	4.30 × 10^−5^
Neurological	106	22.2	1.80 × 10^−4^	8.00 × 10^−4^
Cancer	103	21.5	1.70 × 10^−2^	2.80 × 10^−2^
Psych	81	16.9	4.10 × 10^−5^	2.50 × 10^−4^
Unknown	66	13.8	3.10 × 10^−4^	1.10 × 10^−3^
Developmental	60	12.6	1.10 × 10^−3^	3.40 × 10^−3^
Other	58	12.1	1.10 × 10^−2^	1.90 × 10^−2^
Renal	54	11.3	5.70 × 10^−3^	1.10 × 10^−2^
Reproduction	37	7.7	2.10 × 10^−2^	2.90 × 10^−2^
Normal-variation	33	6.9	7.00 × 10^−6^	6.30 × 10^−5^

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
