# Peer review of "Evolutionary Toxicogenomics of the Striped Killifish (Fundulus majalis) in the New Bedford Harbor (Massachusetts, USA)"

_ijms, 2019, doi:10.3390/ijms20051129_

Reviewer 1 Report

Ruggeri et al. reported analyses of 4000 genome wide SNPs from striped killifish exposed to a gradient of PCBs and other aromatic pollutants in the New Bedford Harbor. The striped killifish populations showed a lack of population differentiation, however they identified 582 candidate outliers along the pollution gradient. Part of these SNPs were found in genes likely associated with the pollutants that occur in NBH. This study is well designed, and the results are very interesting. But, I have one major question about the method for annotation, that was not clearly described in the method section. If authors properly solve the problem in the method of annotation, this manuscript is worthy for publication.

Major comment,

The SNPs were found from the hiseq2500 75 bp single-end sequencing of restriction enzyme fragments. This means the SNPs are in 75 bp DNA fragments. The sequenced fragments were genome-wide, and most of them were from the intergenic regions large distance from genic regions. However, they found annotation (similarity with genes) of 280 SNPs loci (280 fragments) out of 582. It seems too much, because most of the 75 bp fragments may be originated from intergenic regions. Even if the 75 bp sequences are originated from genic regions, the query sequences were only 75 bp, and it is difficult to find high similarity to distantly related human database by blastn search. Please clarify these points.

Minor comments

Please show the map (figure 4) with table 1.

Line 23, NGH should be “New Bedford Harbor (NBH)#.

In introduction section, please provide the information of the life style of striped killifish, especially migration or dispersal.

Line 103, as for “a reference genome”, what does it mean?

Line 210, “sequence tags”, does it mean 75 bp sequences?

Table 3, which part of the gene listed in table 3 were similar to sequence tags? Coding regions? If the sequences were coding regions, blastx search is better.

Line 285, “striped killifish daily movements”, could you explain the detail?

Author Response

We wish to thanks the Reviewer 1 for providing useful comments to our manuscript and contribute significantly to improve it. We hope that the reviewer will find exhaustive how we modified the manuscript under his/her suggestions and the reply to the comments provided here below.

Best regards

Paolo, Xiao, Doug and Margie

____________________________________________________________________________

Reviewer 1

Ruggeri et al. reported analyses of 4000 genome wide SNPs from striped killifish exposed to a gradient of PCBs and other aromatic pollutants in the New Bedford Harbor. The striped killifish populations showed a lack of population differentiation, however they identified 582 candidate outliers along the pollution gradient. Part of these SNPs were found in genes likely associated with the pollutants that occur in NBH. This study is well designed, and the results are very interesting. But, I have one major question about the method for annotation, that was not clearly described in the method section. If authors properly solve the problem in the method of annotation, this manuscript is worthy for publication.

Major comment,

The SNPs were found from the hiseq2500 75 bp single-end sequencing of restriction enzyme fragments. This means the SNPs are in 75 bp DNA fragments. The sequenced fragments were genome-wide, and most of them were from the intergenic regions large distance from genic regions. However, they found annotation (similarity with genes) of 280 SNPs loci (280 fragments) out of 582. It seems too much, because most of the 75 bp fragments may be originated from intergenic regions. Even if the 75 bp sequences are originated from genic regions, the query sequences were only 75 bp, and it is difficult to find high similarity to distantly related human database by blastn search. Please clarify these points.

R: As correctly explained the GBS procedure produces genome–wide DNA fragments associated with a restriction enzyme shearing pattern. These DNA fragments are not necessarily associated with exclusively intergenic portions of the genome but can be part of the coding gene itself. As described in the methods (lines 336-352), candidate outliers were queried against any available GenBank resource using blastn, not just the human database. 237 SNPs out of 539 (28.01%) had hits with significant (E-value < 0.0001) annotations.  Of these 237 SNPs, 151 were related to coding regions (functionally annotated SNPs). NCBI ID codes for these functionally annotated SNPs were converted into human UNIPROT ID codes and used for the enrichment analysis in DAVID 6.7. Reasoning for our approach is listed below:

1-     Put on BLAST-n the 75 bp sequences of the reads associated with each SNPs locus found by the outlier detection methods. The search on BLAST-n was set as “looking for somewhat similar” sequences in Genbank. This choice was set because we are aware that Fundulus majalis is a non-model species and we might have expected very few matching resources. The threshold of an e-value < 0.001 was set in order to discriminate only the more closely related sequence alignments.

2-     The sequences with a significant match were then categorized as “functional” when the description of the match was suggesting a gene, and “non-functional”, in the case of SSRs and other not well specified genomic portions with a clear functional role.

3-     We then took the ID codes provided by NCBI for the list of “functional annotated genes” with potential similarity with the F. majalis outliers loci and we converted these NCBI ID codes into human UNIPROT ID codes. This operation was done for two reasons: a) allow to retrieve the all possible genes functions even from species that have no UNIPROT resources and b) allow us to use DAVID for the enrichment analysis cause the software treats only genomic resources from species with a reference genome (i.e. human, mouse, rats, etc…).

 Minor comments

Please show the map (figure 4) with table 1.

 R: Done

Line 23, NGH should be “New Bedford Harbor (NBH)#.

R: We introduced the acronym NBH at line 17.

In introduction section, please provide the information of the life style of striped killifish, especially migration or dispersal.

R: This information is now provided in lines 136-140:

The striped killifish F. majalis shares similar biological and ecological features with F. heteroclitus. Both killifish species have a sympatric range, but unlike the more well-known congener (F. heteroclitus), the striped killifish is i) seldom found high in the upper-inner tidal, ii) infrequently found in fresher waters and iii) has a greater mobility, which allows this fish to move for several Km and have greater population connectivity [37].

Line 103, as for “a reference genome”, what does it mean?

R: The Fundulus heteroclitus genome assembly was used to aligns the F. majalis reads. This is the sister species of F. majalis and we found that the pipeline that uses a reference for the alignment was slightly more robust that the UNEAK pipeline for analyzing GBS data without a reference genome. This is now detailed in lines 165-168:

Illumina GBS sequencing using the TASSEL pipeline with the Fundulus heteroclitus reference genome (https://www.ncbi.nlm.nih.gov/genome/743) recovered ....

 and lines 809-811:

Sequence reads were aligned and putative SNPs were retrieved using the TASSEL 5.0 pipeline [75] with the F. heteroclitus genome as a reference [76; https://www.ncbi.nlm.nih.gov/genome/743].

Line 210, “sequence tags”, does it mean 75 bp sequences? 

R: Yes, exactly. With “sequence tag” we mean the reads that contain and identify each outlier locus found

Table 3, which part of the gene listed in table 3 were similar to sequence tags? Coding regions? If the sequences were coding regions, blastx search is better.

R: The table 3 reports the pathways of the list of genes found associated with the outliers with functional role. In order to produce the enrichment analysis (with DAVID) that led us to get the KEGG pathways reported in table 3, we converted, the ncbi IDs retrieved after aligning in BLAST-n the sequence tags of the functional loci cointaining an outliers SNPs to human UNIPROT ID codes. Hence, for the enrichment analysis the whole human gene information was used. We can`t, therefore, specify which part of each gene is responsible for the variation observed because the enrichment analysis only identified which physiological pathways and functional clusters of the gene potentially associated with our outliers.  

Line 285, “striped killifish daily movements”, could you explain the detail?

R: we refers to the diel movement ability of striped killifish that can cover distances of Km along the coastline following the planktonic organisms they feed on .

Reviewer 2 Report

Using a GBS dataset with >4000 SNPs, the authors studied patterns of population structure among four samples of killifish collected in the New Bedford Harbor area. Using various outlier detection methods, the authors studied genetic responses to pollution along a “gradient”. Overall, the authors found a staggering amount of more or less robust outliers, which they try to annotate and discuss in great detail these annotations. Overall, I think this makes an interesting contribution to the International Journal of Molecular Sciences, however, there are many small and some larger issues that need to be addressed.

Main issues:

This manuscript is long, often repetitive and the results are redundant with the tables provided (see list of issues below). Abbreviations are often not given when they are used for the first time (e.g. Line 26) and not used in a consistent way (e.g. Line 274). Those are details, but there are many. The methods as described in the results section are often not clear and the conclusion section is quite lost after the methods. To improve readability, please rearrange to a Introduction/Methods/Results/Discussion/Conclusion scheme.

An issue is that the authors picked four sites, sampled fish and then treat those samples as populations. However, is this valid? Given the lack of strong population structure and the very subtle differences presented in Table 1, these could just be part of a large population. This would be consistent with the absence of differentiation and the high mobility of this species as outlined in the discussion. Throughout the manuscript, refer to sites rather than populations. Also, it would be worthwhile to run Structure using only outlier loci (which has not been done in Figure 2) and use these genetic clusters as populations. At least the Structure run using outliers needs to be added.

An other issue are the methods to detect outliers and the number of outliers – they are with almost 600 SNPs very high. This would imply very strong selection and high linkage disequilibrium or a biased representation of GBS SNPs. The authors then go on to suggest that the obtained number of SNPs are actually a conservative estimate (Line 334) also by citing two papers that are more than 10 years old (Line 307) – there were more studies in since then. A crux is that each of the outlier detection method uses different thresholds to account for false positives – in one case, the regression analysis, the authors did not use any such threshold. Also, given that only four sites were used, linear models should not be used. I would suggest to remove the regression method and focus on the three others.

The authors use the site MAT as control because of its distance from the center of pollution – yet the level of PCBs (which was only indirectly extrapolated) are higher here than at the HST site. If there would be a gradient imposing selection, FST estimates using outliers should show this, i.e. FST between HST and MAT should be 0 or very small, and FST between MAT and FAH intermediate and MAT and PIL the highest – this is not the case (Table 2). Also, given the very low level of genetic differentiation and because the authors sampled only fish once from a given site, it remains unclear to which degree this level of differentiation can persist.

Line 18: Remove “evolutionary” – only allelic shifts were studied and these only once at a given site.

Line 21-22: Is there any good reason to assume that the sites are populations?

Line 26: NBH not introduced before.

Line 32: Replace “the major threat” with “a major threat”.

Line 41: Is there any evidence for mass-mortality? Could you calculate catch-per-unit-efforts from your sampling? The genetic results in Table 1 speak against mass-mortality.

Line 58: What is an US Superfund?

Line 63: PCB not introduced before.

Line 74: This is a very large population size, but how are populations delineated in this species?

Line 95: Replace “paper” with “study”

Line 95: Replace “evolutionary” with “population genetic” or just “genetic” – this seems to be more a case for selection from standing genetic variation.

Line 116 onwards: Remove the text that is redundant with the table.

Line 139: DAPC not introduced before. Also add the % of variance explained.

Figure 1: Indicate % of variance explained by each axis in the DAPCs – the bar plot with the eigenvalues misses a bar. Also, move the population labels outside the confidence circle to increase readability. Add a Structure plot using only outlier loci.

Line 155: Numbers are odd.

Line 157: Replace ln probability with log likelihood.

Figure 2: Make arrowheads larger.

Line 167: Now you speak about “sampling sites” rather than populations.

Line 200: What is a “replicate” in this case? The BayesFactor cut-offs seem to be rather low, what is the good practice for this? Alternatively, a hidden markov model could be run on the BayesFactors to identify the statistical outliers.

Line 204: write “potential candidate SNP outliers”.

Line 217: The SNPs are not functional unless you tested for non-synonymous effects.

Line 244: remove “we investigated how GBS can help to clarify”

Line 247: The loci did not evolve unless you expect de novo mutations, you only study allelic shifts as a result of drift and selection.

Line 258: A pity though that you do not have any phenotypic data to support any of these claims.

Line 265: But there is no evidence for such an erosion.

Line 272-273: This is wrong – there is no difference – check the 95% CI in Table 1.

Line 288: Replace “evolutionary” with “genetic”.

Line 298: Remove “important”

Line 307: The cited studies are 10 years old!

Line 313: Again, what is the rational to call them populations?

Line 322-334: This part should be more balanced given that you have too many rather too few outliers.

Line 345: It is “functionally annotated SNPs”

Line 346: The traits are not important but they might be important in the response to pollution. Also, given the Venn diagram, perhaps it should be indicated which of the SNPs and annotations was found by all, some or only one outlier method. This would identify the sites that are more likely under selection.

Line 352: These are not functional.

Lines 413-415: Again, the shortcoming is not that there are too few but too many outliers.

Line 449: More details about the reference genome should be provided (N50 aso).

Line 479: Indicate the total % explained by these PCs.

Line 500: How was this done in this study? Did you use the negative FSTs, zero values? Many of the approximations used to run Migrate seem to be violated by the shallow populations structure among sampling sites.

Line 510: How accurate are the extrapolated values? And how stable are they through time?

Line 510: Still, a linear regression with only 4 samples is statistically very limited. Remove.

Line 534: Logically, how can you remove outlier loci from an analysis that identifies outliers before you ran it?

Line 545: Check NCBI for the citation of blastn.

Line 565 onwards: Better discuss this in the light of the limitations of the outlier methods used when population structure is very shallow as here.

Line 571: It is good practice in the field to deposit the raw sequence data on NCBI Short Read Archive or the European Nucleotide Archive. This is a must for publication.

Author Response

We wish to thanks the Reviewer 2 for providing useful comments to our manuscript and contribute significantly to improve it. We hope that the reviewer will find exhaustive how we modified the manuscript under his/her suggestions and the reply to the comments provided here below.

Best regards

Paolo, Xiao, Doug and Margie

____________________________________________________________________________

Reviewer 2

Comments and Suggestions for Authors

Using a GBS dataset with >4000 SNPs, the authors studied patterns of population structure among four samples of killifish collected in the New Bedford Harbor area. Using various outlier detection methods, the authors studied genetic responses to pollution along a “gradient”. Overall, the authors found a staggering amount of more or less robust outliers, which they try to annotate and discuss in great detail these annotations. Overall, I think this makes an interesting contribution to the International Journal of Molecular Sciences, however, there are many small and some larger issues that need to be addressed.

Main issues:

This manuscript is long, often repetitive and the results are redundant with the tables provided (see list of issues below). Abbreviations are often not given when they are used for the first time (e.g. Line 26) and not used in a consistent way (e.g. Line 274). Those are details, but there are many. The methods as described in the results section are often not clear and the conclusion section is quite lost after the methods. To improve readability, please rearrange to a Introduction/Methods/Results/Discussion/Conclusion scheme.

R: The idea to change the order of sections as proposed was appreciated but after confirmation from the Editor we have to maintain the original format (as requested by IJMS). We also checked carefully for acronyms used throughout the manuscript.

An issue is that the authors picked four sites, sampled fish and then treat those samples as populations. However, is this valid? Given the lack of strong population structure and the very subtle differences presented in Table 1, these could just be part of a large population.

R: We are sorry for the misleading word “population” that we sloppily used as a synonym for “sample”. We are aware that the individuals collected for this paper are genetically part of the same population (as already observed from DAPC and STRUCTURE). Throughout the manuscript we have changed the word “population” to “sample or sampling site” in order to avoid any further confusion.

This would be consistent with the absence of differentiation and the high mobility of this species as outlined in the discussion. Throughout the manuscript, refer to sites rather than populations. Also, it would be worthwhile to run Structure using only outlier loci (which has not been done in Figure 2) and use these genetic clusters as populations. At least the Structure run using outliers needs to be added.

An other issue are the methods to detect outliers and the number of outliers – they are with almost 600 SNPs very high. This would imply very strong selection and high linkage disequilibrium or a biased representation of GBS SNPs. The authors then go on to suggest that the obtained number of SNPs are actually a conservative estimate (Line 334) also by citing two papers that are more than 10 years old (Line 307) – there were more studies in since then. A crux is that each of the outlier detection method uses different thresholds to account for false positives – in one case, the regression analysis, the authors did not use any such threshold. Also, given that only four sites were used, linear models should not be used. I would suggest to remove the regression method and focus on the three others.

R: We agree with the Reviewer and the regression analysis was removed from the set of analysis performed to look for outliers. All the following analysis (i.e. count of outliers, enrichment analysis, etc. were repeated without outliers identified with the regression method).

The authors use the site MAT as control because of its distance from the center of pollution – yet the level of PCBs (which was only indirectly extrapolated) are higher here than at the HST site. If there would be a gradient imposing selection, FST estimates using outliers should show this, i.e. FST between HST and MAT should be 0 or very small, and FST between MAT and FAH intermediate and MAT and PIL the highest – this is not the case (Table 2). Also, given the very low level of genetic differentiation and because the authors sampled only fish once from a given site, it remains unclear to which degree this level of differentiation can persist.

R: we defined MAT as a control because it is the farthest sample we have from the New Bedford Harbor (MAT belong to a different embayment) and hence from the source of pollution. The MAT and HST have similar values in PCBs and we guessed that the local pressure imposed by pollutants on these samples are much less than what experiencing samples in PIL and FAH sites. 

Line 18: Remove “evolutionary” – only allelic shifts were studied and these only once at a given site.

R: Done.

Line 21-22: Is there any good reason to assume that the sites are populations?

R: any referring to “populations” was removed or changed with “sampling sites”.

Line 26: NBH not introduced before.

R: we now introduced the acronym at line 17.

Line 32: Replace “the major threat” with “a major threat”.

R: done.

Line 41: Is there any evidence for mass-mortality? Could you calculate catch-per-unit-efforts from your sampling? The genetic results in Table 1 speak against mass-mortality.

R: In the introduction we explain what usually happens with acute severe pollution in aquatic systems. This was also the case of New Beford Harbor from 1940s till 1970s when there was a huge release of toxicants from the River Acushnet. Since then the PCBs, heavy metals and other aromatic compounds in the area are still maintained at very high (in most of the case lethal concentrations) values, and the pollution in the area became chronic. There was a big reduction in the species number over this amount of time and among the few fish species Fundulus heteroclitus and F. majalis are the more common species found throughout the year. We don`t have any estimation of mass-mortality for F. majalis, and this was not our aim in the context of this study. We hope that with the above explanation we clarified and focused our point of view concerning the topics reported in the sentence at lines 42-45.

Line 58: What is an US Superfund?

R: A US Superfund site is a polluted sited designated for cleanup with funds from the responsible party. This is now defined in the manuscript (lines 94-95).

Line 63: PCB not introduced before.

R: Now introduced at line 16

Line 74: This is a very large population size, but how are populations delineated in this species?

R: The population size information as the other biological information were provided for the sister species Fundulus heteroclitus that is a model species for ecotoxicology and way much better studied that the species used for this study (F. majalis). We current have no precise information about the population size of F. majalis but the two species have an overlapping range distribution and similar population densities all throughout their range of occurrence.

Line 95: Replace “paper” with “study”

R: done

Line 95: Replace “evolutionary” with “population genetic” or just “genetic” – this seems to be more a case for selection from standing genetic variation.

R: done.

Line 116 onwards: Remove the text that is redundant with the table.

R: done

Line 139: DAPC not introduced before. Also add the % of variance explained.

R: done

Figure 1: Indicate % of variance explained by each axis in the DAPCs – the bar plot with the eigenvalues misses a bar. Also, move the population labels outside the confidence circle to increase readability. Add a Structure plot using only outlier loci.

R: done

Line 155: Numbers are odd.

There was a mistake with the writing of the number of clusters with only neutral SNPs (K=2 + K=4) and the whole dataset (K=3). The sentence was checked and rewritten for clarity.

Line 157: Replace ln probability with log likelihood.

R: done

Line 167: Now you speak about “sampling sites” rather than populations.

R: We are sorry for the confusion concerning the usage of the term “population” as synonym of “sampling site”. We checked the manuscript to avoid any further confusion or ambiguity.

Line 200: What is a “replicate” in this case? The BayesFactor cut-offs seem to be rather low, what is the good practice for this? Alternatively, a hidden markov model could be run on the BayesFactors to identify the statistical outliers.

R: We used the Bayes factor cut-off suggested by Bayenv 2 and applied also in other published papers that used Bayenv 2. To maximize the probsbility of finding true candidate outliers we repeated the runs in Bayenv 2 multiple times and looked at loci that emerged as outliers consistently among the repeated analysis.

Line 204: write “potential candidate SNP outliers”.

R: done

Line 217: The SNPs are not functional unless you tested for non-synonymous effects.

R: In this context we wanted to suggest the SNPs that were found associated with a coding gene. We now use the manuscript the code “functionally annotated SNPs” as you suggested in a comment below.

Line 244: remove “we investigated how GBS can help to clarify”

R: Done

Line 247: The loci did not evolve unless you expect de novo mutations, you only study allelic shifts as a result of drift and selection.

R: the sentence was modified accordingly with the suggestion proposed above.

Line 258: A pity though that you do not have any phenotypic data to support any of these claims.

R: Agreed. Phenotypic data that correlates well with genotypic data is ideal. In our case, we can only suggest that these loci might allow F. majalis to inhabit the polluted NBH sites.  

Line 265: But there is no evidence for such an erosion.

R: Here we are simply contrasting our results with the expectations from the genetic erosion hypothesis. This theory suggests that a progressive decline in genetic variation should be observed when facing with an increase in pollutants. This was not the case we found.

Line 272-273: This is wrong – there is no difference – check the 95% CI in Table 1.

The sentence was just a general observation that is not supported by our results, and it was removed from the manuscript.

Line 288: Replace “evolutionary” with “genetic”.

R: done

Line 298: Remove “important”

R; done

Line 307: The cited studies are 10 years old!

R: The eldest of the cited studies is from 2012.

Line 313: Again, what is the rational to call them populations?

R: the sentence was modified accordingly with the reviewer`s comment.

Line 322-334: This part should be more balanced given that you have too many rather too few outliers.

R: the sentence was changed as requested.

Line 345: It is “functionally annotated SNPs”

R: done

Line 346: The traits are not important but they might be important in the response to pollution. Also, given the Venn diagram, perhaps it should be indicated which of the SNPs and annotations was found by all, some or only one outlier method. This would identify the sites that are more likely under selection.

R: The functional annotated loci with overlapping between methods to detect outliers are:

LBHS9887_384963Labrus bergylta SWI/SNF related, matrix   associated, actin dependent regulator of chromatin, subfamily d, member 3   (smarcd3), transcript variant X1, mRNA
LBS10017_3443539Larimichthys crocea gamma-aminobutyric acid receptor subunit alpha-5   (LOC104929488), transcript variant X1, mRNA

S1327_21414Fundulus heteroclitus sphingosine-1-phosphate phosphatase 1-like   (LOC105935524), mRNA

S243_328842Fundulus heteroclitus golgin subfamily A member 6-like protein 22   (LOC110368556), transcript variant X4, misc_RNA

S243_328842Fundulus heteroclitus filamin A interacting protein 1 like (filip1l),   transcript variant X8, mRNA

S9048_1026Fundulus heteroclitus carcinoembryonic antigen-related cell adhesion   molecule 1-like (LOC110367333), mRNA

S9884_420423Cyprinodon variegatus centrosomal protein 63kDa (cep63), mRNA

S9899_1841030Fundulus heteroclitus syntaxin-2 (LOC105928768), transcript variant X2,   mRNA

S9975_4378024Fundulus heteroclitus nuclear receptor subfamily 2 group C member 2   (nr2c2), mRNA
LHS42_249281 Fundulus heteroclitus thyrotropin   releasing hormone (trh), mRNA

S843_122311Fundulus heteroclitus mannose receptor C-type 1 (mrc1), transcript   variant X1, mRNA

LBH= the all three methods

LB= lositan+Bayenv

LH= Lositan+HIM

The other overlapping loci contained no functionally annotated SNPs. We discussed these genes and their role into the discussion section dedicated (3.3. Genome-environment interaction in F. majalis from NBH) and we added in the Table S3 the method used to detect any functionally annotated SNPs.

Line 352: These are not functional.

R: We don`t understand which is the subject of the request. At the line suggested we are mentioning only the Synthaxin-2 that is a gene for sure with a coding activity into the genome.

Lines 413-415: Again, the shortcoming is not that there are too few but too many outliers.

R: In the associated lines (if we picked the right ones) we are just explaining that the GBS method is usually non-comprehensive in scanning the whole genome and this can lead to (sometimes) not targeting other relevant genes connected with an environmental response. We are therefore not saying that we were expecting more outliers loci but just explaining that other loci emerged in analogous papers might have found a different set of loci because of the above mentioned drawback. We are willing to remove this part of the discussion if the reviewer feels that it is suggesting a contradiction with what expressed before in the manuscript.

Line 449: More details about the reference genome should be provided (N50 aso).

An hyperlink that remind to the information of the Fundulus heteroclitus genome in genebank was added and now it is viable to anyone want to consult it.

Line 479: Indicate the total % explained by these PCs.

R: we added the % of variance explained by PC1 and PC2 in the result section.

Line 500: How was this done in this study? Did you use the negative FSTs, zero values? Many of the approximations used to run Migrate seem to be violated by the shallow populations structure among sampling sites.

R: I`m not sure you can superimpose the choice of any threshold in the FSTs that are retained and used by MIGRATE-n. There is only the possibility to use the FSTs or other genetic distance measures during the process of setting MIGRATE-n. We decided to use FSTs as a metrics for genetic relatedness, as commonly reported in many papers. We looked also at graphs that the software produces to estimate the degree of confidence in the CI and likelihood obtained, and all them showed the expected hump-shape profile.

We aimed at using MIGRATE-n as an additional supporting method to the assignment test in geneclass.   

Line 510: How accurate are the extrapolated values? And how stable are they through time?

This part of the manuscript was deleted after the suggestion to remove the regression analysis performed to look at outliers associated with a linear regressing pattern.

Line 510: Still, a linear regression with only 4 samples is statistically very limited. Remove.

R: done

Line 534: Logically, how can you remove outlier loci from an analysis that identifies outliers before you ran it?

R: We see your point but for running Bayenv 2 one important step is the removal of any locus that can have a genetic linkage and that can be under non-neutral variation because of linkage disequilibrium. We therefore needed to prepare a dataset containing exclusively loci that are not in linkage and that are therefore considered as a “presumed neutral” baseline.

Line 545: Check NCBI for the citation of blastn.

R: citation added

Line 571: It is good practice in the field to deposit the raw sequence data on NCBI Short Read Archive or the European Nucleotide Archive. This is a must for publication.

R: The genomic resources produced by this study will be uploaded consequently after the publication of the manuscript in an International public repository as already declared during the submission process.

Round  2

Reviewer 1 Report

I agree with all modifications in the revised manuscript.

Now, the manuscript is improved and ready for publication.

Author Response

We wish to thanks a lot the Review 1 for the great contribution to the improvement of our manuscript.

Best Regards

Paolo, Xiao, Doug and Margie

Reviewer 2 Report

The authors have greatly improved the manuscript, making it more clear and readable. However, I still have two major concerns:

First: Please check that the structure runs reported in Figure 1 are correct – the not-neutral and neutral loci based runs are identical except that the colors were flipped. This cannot be unless you would suggest that the adaptive loci yield the same population structure as the non-adaptive - in which case they would not be adaptive (or neutral despite being "functional" by being in genes).

The second is still the Migrate-n analysis. Did you set the negative FST values to zero as other studies do (see e.g. Riley et al. 2016 Bot J Linn Soc, or Poesel et al. 2017 Ethology). If not, please make a good case why not. Check also Roesti et al. 2012 BMC Evol Biol for a discussion about the value of negative FSTs.

A last minor issue:

Figure 1: The axis labels should be DAPC and not PC. Either provide an y axis for the DA eigenvalues or remove this inlet (can be easily done in Adobe Illustrator or Inkscape).

Author Response

We want to thanks again the Reviewer 2 for the great help provided to ameliorate the quality of this manuscript.

Here below the reply point by point:

The authors have greatly improved the manuscript, making it more clear and readable. However, I still have two major concerns:

First: Please check that the structure runs reported in Figure 1 are correct – the not-neutral and neutral loci based runs are identical except that the colors were flipped. This cannot be unless you would suggest that the adaptive loci yield the same population structure as the non-adaptive - in which case they would not be adaptive (or neutral despite being "functional" by being in genes).

R: We are terribly sorry but a wrong plot was included for the NON-Neutral Structure outcome. In the effort of providing promptly by the deadline expected by the Journal we picked the wrong plot that was actually another Structure run with neutral loci. We fixed now this issue and we apologize for the mistake.

The second is still the Migrate-n analysis. Did you set the negative FST values to zero as other studies do (see e.g. Riley et al. 2016 Bot J Linn Soc, or Poesel et al. 2017 Ethology). If not, please make a good case why not. Check also Roesti et al. 2012 BMC Evol Biol for a discussion about the value of negative FSTs.

R: Yes we set zero as Fst baseline for Migrate-n, we apologize if we gave this as took for granted. We reported this setting in the material and method section at line 488.

A last minor issue:

Figure 1: The axis labels should be DAPC and not PC. Either provide an y axis for the DA eigenvalues or remove this inlet (can be easily done in Adobe Illustrator or Inkscape).

R: for simplicity the inlet eigenvalues plot was removed from the figure.